# Humoral Response after Two Doses of BNT162b2 mRNA Vaccine Has a Role in Predicting Response after Three Doses That Is Related to Plasma HIV Viremia and Nadir CD4+ Cell Count in HIV-Positive Patients

**DOI:** 10.3390/vaccines11010082

**Published:** 2022-12-30

**Authors:** Monica Basso, Nicole Pirola, Susanna Pascoli, Beatrice Bragato, Antonio Vinci, Marco Iannetta, Francesco Colombo, Nicholas Geremia, Luca Martignago, Maria Cristina Rossi, Ludovica Cipriani, Mario Giobbia, Pier Giorgio Scotton, Saverio Giuseppe Parisi

**Affiliations:** 1Department of Molecular Medicine, University of Padova, Via Gabelli, 63, 35100 Padova, Italy; 2Microbiology Unit, Department of Specialist and Laboratory Medicine, Ca’ Foncello University Hospital, 31100 Treviso, Italy; 3Hospital Health Management Area, Local Health Authority “Roma 1”, Borgo Santo Spirito 3, 00193 Roma, Italy; 4Infectious Disease Unit, Department of System Medicine, Tor Vergata University and Hospital, Via Montpellier 1, 00133 Rome, Italy; 5Unit of Infectious Diseases, Department of Clinical Medicine, Ospedale dell’Angelo, Via Paccagnella, 11, 30174 Venice, Italy; 6Infectious Diseases Unit, Treviso Hospital, Via Antonio Scarpa, 2, 31100 Treviso, Italy

**Keywords:** HIV infection, BNT162b2 mRNA vaccine, third dose, nadir, HIV residual viremia, trimeric S-protein antigen, IgG

## Abstract

We investigated the spike IgG levels of HIV+ patients on antiretroviral therapy six months after they received their second dose (T2) and six months after the third dose (T3) of the BNT162b2 mRNA vaccine, as well as the influence of different levels of plasma HIV viremia of overall CD4+ cell count and nadir value on the humoral time course. One hundred eighty-four patients were enrolled. The median age was 55 years, the median CD4+ cell count was 639 cells/mm^3^ and the median nadir value was 258 cells/mm^3^. On the basis of all tests performed during the study period, persistently undetectable plasma HIV RNA (PUD) was found in 66 patients, low-level viremia (LLV) in 57 and ongoing viremia (OV) in 61. Serum levels of IgG antibodies against a trimeric S-protein antigen were tested with DiaSorin Liaison SARS-CoV-2 TrimericS IgG and the response was classified as optimal (>75th percentile), intermediate (50th–25th percentile) and low (<25th percentile). The frequencies of the three different patterns of plasma HIV viremia (PUD, LLV and OV) were comparable in patients with low, intermediate and optimal IgG response evaluated at T2, with no difference in overall CD4+ cell count or nadir count. At T3, 92.9% of patients achieved an optimal response: T2 response proved to be the most important factor in predicting T3 optimal response in patients with LLV and OV.A nadir value ≤ 330 cells/mm^3^ had 100% sensitivity in predicting a non-optimal response. In conclusion, we demonstrated the persistence of anti-spike IgG, with high serum levels occurring in most patients six months after the third dose of the BNT162b2 mRNA vaccine and a predictive role of humoral response at T2 in subjects with detectable plasma HIV viremia. Immunological alterations related to past immunodeficiency may persist despite immune reconstitution, and the nadir value could be a useful tool for elaborating personalized vaccine schedules.

## 1. Introduction

For three years, the world has been challenged by the coronavirus disease 2019 (COVID-19) pandemic, and a tremendous global effort has been made to tackle it. At the same time, we face other major health threats, such as infection with the human immunodeficiency virus (HIV), which is a disease affecting approximately 38 million people [1,2].

The complexity of HIV infection may explain its multifaceted influence on SARS-CoV-2 infection outcome. HIV infection is an independent risk factor for both severe COVID-19 at admission and in-hospital mortality [3], and it is associated with a higher risk of severe disease due to SARS-CoV-2 infection [4], impaired immunity (low CD4+ cell number defined both as <200 cells/mm^3^ and as <350 cells/mm^3^) [5], CD4 + T-cell count nadir <200 cells mm^3^ [6] and the coexistence of comorbidities known as risk factors (i.e., diabetes and cardiovascular diseases), which are frequently diagnosed in HIV-positive patients [7] and are all factors involved. There is also an interplay between SARS-CoV-2 infection and response to antiretroviral therapy; viral suppression (defined as HIV RNA < 200 copies/mL) is negatively associated with hospitalization [8], and the risk of a severe outcome in patients with a CD4+ cell count <200 cells/mm^3^ is described only in the case of detectable plasma HIV viremia (>50 copies/mL) [9]. Conversely, no association has been found between COVID-19 severity and viremia (classified as 0, 1 or 2 HIV RNA values <50 copies in the 6 months before SARS-CoV-2 infection) [10]. Taken together, these data point out the importance of COVID-19 vaccination in HIV-infected patients and indicate an interest in characterizing the possible influence of ongoing HIV replication, even at a low level, on long-term humoral response to COVID-19 vaccination. It is known that patients with unsuppressed HIV viremia experience impaired protection after live vaccine administration, and a rapid vaccine schedule (i.e., for tick-borne encephalitis vaccine) should not be applied because of the risk of lower seroconversion rates [11,12]. Residual HIV viremia is a complex phenomenon, and no univocal threshold of HIV RNA copies/mL has been identified. Underlying mechanisms are poorly understood, and many hypotheses have been reported, including monocyte/macrophage activation, reactivated infections and interplay between HIV infection and comorbidities [13,14,15].

The aims of the study were to evaluate the time course of the response of IgG antibodies to SARS-CoV-2 spike protein six months after the second and third vaccine doses in HIV patients in a real-life setting and to describe possible correlations with CD4 + cell count and plasma HIV viremia.

## 2. Materials and Methods

### 2.1. Study Design

Eligible patients were HIV-1-positive adults on successful antiretroviral therapy (ART) who had been vaccinated with three doses of the BNT162b2 mRNA vaccine. Individuals with a prior laboratory-confirmed COVID-19 or anamnestic data suggesting a possible SARS-CoV-2 infection not further evaluated were excluded. No other exclusion criteria were applied.

Parameters evaluated at enrollment (before the first vaccine dose) were gender, age, age at the diagnosis of HIV infection, CD4+ cell count at nadir (the person’s lowest CD4+ cell count) and ongoing ART.

CD4+ cell count and plasma HIV viremia were evaluated at three study points: before the first vaccine dose, 6 months after the second vaccine dose (T2) and 6 months after the third vaccine dose (T3). CD4+ cell count was analyzed using two variables; the first corresponded to the mean CD4+ count values evaluated before the first vaccine dose, at T2 and at T3 and it was defined overall CD4+ cell count, and the second variable corresponded to the difference between overall CD4+ cell count and CD4+ cell count at nadir, and it was called delta CD4+ count.

Patients were classified as having persistently undetectable (PUD) plasma HIV viremia if the plasma value was undetectable in all tests performed; low-level viraemia (LLV) if no plasma HIV-RNA value was >20 copies/mL; and ongoing viremia (OV) if at least one plasma value was >20 copies/mL. In this last category of patients, the occurrence of blips (isolated viral loads (VLs): 50–499 copies/mL between measurements <50 copies/mL) was also evaluated [16].

The study received approval from the Comitato per la Sperimentazione Clinica di Treviso e Belluno (protocol code 812/2020), and written informed consent was obtained from all the patients enrolled.

### 2.2. SARS-CoV-2 Serology

Serum levels of IgG antibodies against a trimeric S-protein antigen were tested 6 months after the second and third vaccine dose with DiaSorin Liaison SARS-CoV-2 TrimericS IgG (DiaSorin Inc, Stillwater, MN, USA), an indirect chemiluminescence immunoassay, in accordance with the manufacturer’s instructions [17].

The IgG titers were expressed in binding antibody units (BAUs). Samples with values < 33.8 BAU/mL were considered negative. The lower limit of detection was 4.81 BAU/mL, which was scored as 2 for statistical analysis, while a value > 2080 BAU/mL (the higher limit of quantification) was scored as 2080 for statistical analysis.

The patients were classified according to IgG value after the second vaccine dose, as follows: optimal responders (IgG value higher than 75th percentile), intermediate responders (IgG value ranging from 50th to 25th percentile) and low responders (IgG value lower than 25th percentile). The value limit applied for the second dose was also applied to the serum IgG values tested six months after the third vaccine dose.

### 2.3. Statistical Analysis

Continuous variables (age at HIV infection, age at study enrollment, CD4+ cell count at nadir, overall CD4+ cell count and delta CD4+ cell count) were evaluated as the median and interquartile ranges. The Mann–Whitney U test was used to test the significance of differences in the median between groups of patients. Differences in proportions were calculated with the chi-squared test and Fisher’s exact test as appropriate. Sensitivity and specificity of continuous variables were determined by receiver operating characteristic curve analysis. Among the categorical variables, IgG responses at T2 and T3 (as defined in Section 2.2) were treated as ordinal variables, and optimal responsiveness at T2 and T3 were treated as binary (dummy) variables. Univariate logistic analysis has been performed on the candidate predictors [18] in order to elaborate an overall multivariate model. The limit for statistical significance in all analyses was established at *p* < 0.05. Statistical analyses were performed with MedCalc^®^ Statistical Software version 20.111 (MedCalc Software Ltd., Ostend, Belgium; https://www.medcalc.org; 2022) and with Stata v. 17.0. (StataCorp. College Station, TX, USA; https://www.stata.com; 2021).

## 3. Results

One hundred eighty-four patients were included in the study. The median age at enrollment was 55 years and the median overall CD4+ cell count was 639 cells/mm^3^ (IQR 486–845 cells/mm^3^). A total of 118 subjects (64.1%) were on triple therapy. HIV viremia control was equally distributed in the study population (66 patients with PUD, 57 with LLV and 61 patients with OV). In total, 15 (24.6%) of the 61 patients with OV had at least 1 plasma HIV RNA value > 50 copies/mL, while only 4 subjects (6.6%) had a value higher than 100 copies/mL. Viral blips occurred in 18 subjects (9.8% of the overall study population). The frequency of patients with an IgG serum level > 2080 BAU/mL was significantly higher at T3 (64.1% vs. 8.7%, *p* < 0.0001). A complete description is reported in Table 1.

Seroconversion was achieved in almost all patients at T2 and at T3 (181 and 183, respectively). Of note, all three subjects negative at T2 had an overall CD4+ cell count lower than 500 cells/mm^3^ (6.3% of the 48 subjects had a CD4+ cell count lower than 500 cells/mm^3^ vs. none of the 136 patients with a CD4+ cell count higher than 500 cells/mm^3^, *p* = 0.016934).

### 3.1. Characteristics of Humoral Response at T2

At T2, the higher limit for the first quartile was 148 BAU/mL and the lower limit for the fourth quartile was 651 BAU/mL. Median values and IQR were 102 BAU/mL (69–119 BAU/mL) in low responders, 285 BAU/mL (228–418 BAU/mL) in intermediate responders and 1650 BAU/mL (1000–2080 BAU/mL) in optimal responders. 

The frequencies of the three different patterns of plasma HIV viremia (PUD, LLV and OV) were comparable in patients with low, intermediate and optimal IgG responses at T2. No difference in median CD4 cell count at nadir or in median overall CD4 cell count was found between patients with low, intermediate or optimal IgG responses at T2 within the same plasma HIV RNA figure: A median value of overall CD4 lower than 500 cells/mm^3^ was found only in patients with OV and a low response. The data are detailed in Table 2.

Logistic regression analysis was performed to investigate if CD4+ cell count, CD4+ cell count at nadir and the CD4+ delta value could predict IgG response at T2. No predictor emerged in this phase. Data are summarized in Table 3.

### 3.2. Humoral Response at T3 and Longitudinal Evolution 

At T3, most patients achieved an optimal response (171, 92.9%). A low response was found in only four patients, all of whom had a low response at T2 (*p* = 0.011 with respect to none in patients with intermediate response at T2). An optimal response was confirmed at T3 in 100% of the patients with an optimal response at T2, in 95.7% of the patients with an intermediate response at T2 and in only 80.4% in the case of patients with a low response at T2 (0.002536) (Table 4).

Only one patient did not achieve seroconversion at T3; he was a 72-year-old male with a nadir cell count of 34 cells/mm^3^ and an overall CD4+ count of 150 cells/mm^3^ and LLV, and he was on treatment with dolutegravir and lamivudine.

The median value of overall mean CD4 cell count and CD4 count at nadir were lower in the 9 patients with a low response at T2 that did not achieve an optimal response compared to the 37 patients who had succeeded (583 cells/mm^3^, IQR 376–820 cells/mm^3^ vs. 644 cells/mm^3^, IQR 382–1019 cells/mm^3^, *p* = 0.4058 and 178 cells/mm^3^, IQR 91–300 cells/mm^3^ vs. 305 cells/mm^3^, IQR 165–424 cells/mm^3^, *p* = 0.0592, respectively).

Logistic regression analysis was again performed on the same variables as before, with the addition of T2 response as possible predictor. T2 response proved to be the most important factor in predicting T3 optimal response. In some instances, it was observed that patients with low T2 response had still a moderate response on T3, while the opposite was never true, meaning that at T3 patients had either improved or maintained their previous response level. The predictive value of T2 response was more evident in patients with OV and LLV as shown in Table 5. None of the predictors were found to be useful in the multivariate model.

No predictor variable was found among patients who had had a good response to T2; instead, in patients with low response at T2, a CD4+ cell count at nadir ≤ 330 cells/mm^3^ had 100% sensitivity and a specificity of 47.22% in predicting a low or intermediate response at T3 (AUC 0.715, 95% CI 0.559–0.841, *p* = 0.0138); the corresponding logistic model describes the probability of optimal T3 response in these patients. (Figure 1).

At T3, the higher frequency of serum IgG value >2080 BAU/mL (78.3%) was found in patients with an optimal response at T2. Conversely, the lowest frequency was found in patients with a low response at T2 (43.5%), being significantly lower than the frequency in patients with an intermediate response at T2 (43.5% vs. 67.4%, *p* = 0.0072) and an optimal response (43.5% vs. 78.3%, *p* = 0.0007). The relative frequency of an IgG value >2080 BAU/mL with respect to the other IgG values included in the 75th percentile was 54% in subjects with a previously low response, 70.5% in the case of a previously intermediate response and 78.3% when an optimal response was confirmed at T3 (*p* = 0.02 with respect to low response) (Figure 2).

The frequency of patients with PUD, LLV and OV was comparable in patient cohorts with different longitudinal evolutions of IgG response at T3; the numerosity of each group is extremely different, but patients with OV were included in all cohorts with a frequency ranging from 25% to 50% (Figure 3).

## 4. Discussion

In this study, we investigated the spike IgG levels of HIV+ patients on antiretroviral therapy six months after they received their second dose (T2) and six months after the third dose (T3) of the BNT162b2 mRNA vaccine, as well as the influence of different levels of plasma HIV viremia of overall CD4+ cell count and nadir value on the humoral time course. First, we observed high IgG serum levels in 92.9% of patients six months after the third dose, and only one subject confirmed failed seroconversion at T2. A few works reported data for HIV patients receiving mRNA-based vaccines; the efficacy of the third dose was confirmed in all reports, but the study designs differed from ours in terms of patient characteristics, the interval between third dose, and antibody testing. HIV patients had a median CD4+ cell count <200 cell/mm^3^ and/or a previous AIDS diagnosis at the first vaccine dose per protocol in the work of Vergori et al. [19] and a median CD4+ cell count >600 cells/mm^3^ in the other studies [20,21]. The time from the third dose was lower (range 2–8 weeks) [19,20,21], and the humoral response was assessed with anti-S/RBD tests or with antitrimeric spike-protein-specific IgG [19,20].

Previous studies reported no correlation between undetectable plasma HIV viremia, conventionally defined as plasma HIV RNA lower than 20 copies/mL [22] or 30 copies/mL [23], and humoral response, so we planned to separately evaluate the impact of persistent plasma HIV RNA complete suppression, defined as confirmed undetectable plasma HIV RNA versus low level HIV viremia, to show if absence of clinically quantifiable ongoing viremia could influence response to the COVID-19 vaccine.

Interestingly, we observed that serum IgG level at T2 proved to be the most important factor predicting optimal response at T3 in patients with LLV and with OV but not in patients with PUD; this is the second main result of the study. Detectable plasma HIV viremia played a role in counteracting the peak antibody response even in the case of minimal viral burden; this result may be explained by the expansion of functionally exhausted tissue-like memory B cells, which is associated with chronic HIV viremia [24]. These cells have a lower ability to proliferate, to undergo affinity maturation and to secrete antibodies and cytokines [25]. We classified the patients as having a LLV on the basis of a plasma HIV RNA value <20 copies/mL detected during the study period; ultrasensitive (2.5 copies/mL) real-time PCR used to quantify plasma HIV RNA could have been a useful tool for better classifying the patients, and also for the inverse correlation between immune activation and full suppression [26,27,28], but this was beyond the aims of this real-life study.

The interplay between plasma HIV viremia and humoral response was also studied with a focus both on vaccine-related immunological activation and vaccine efficacy; potential vaccine-related viral blips were reported with a frequency of up to about 9% [23,29,30] (comparable to ours). After the second vaccine dose, lower levels of spike IgG in patients with baseline HIV-RNA more >50 copies/mL were described [30], and most subjects with plasma HIV-RNA >200 copies/mL were anti-RBD IgG non-responders [31].

The overall CD4+ cell did not influence evolution of humoral response from T2 to T3. The CD4+ cell count reflects the immunocompetence level [32], and the median value was higher than 500 cells/mm^3^ in all patients except for those with a low response, and at least one HIV RNA plasma value of >20 copies/mL (OV); however, they had a median value of 427 cells/mm^3^, which is more than double the threshold of 200 CD4+ cells/mm^3^ associated with a low humoral response to the COVID-19 vaccine [31,33]. 

Conversely, the CD4 cell count at nadir was lower in subjects with a low response after the second dose, evolving to a low or intermediate response after the following dose (with respect to patients with low and then optimal responses), and a nadir value ≤330 cells/mm^3^ had 100% sensitivity in predicting a non-optimal response after the last vaccine dose. The significance of the nadir value as a factor predicting response to the COVID-19 vaccine is crucial because it expresses past immunodeficiency and could be the only routine clinical data available for this evaluation in high-income countries, where almost all patients have a current CD4 count >500 cells/mm^3^ due to ongoing ART. The generation of a vaccine-induced humoral response requires cooperation between antigen-primed B cells and T-follicular helper cells; advanced HIV disease is associated with alterations in B cells and T-helper cells, and immune reconstitution post-ART would be incomplete with the presence of activated and immature B cells in peripheral blood, apoptotic B cells in lymph nodes and circulating T-helper follicular cells with poor survival and function [34,35]. Nadir was previously studied as a factor with a possible influence on humoral response with negative results. Lapointe et al. [36] found no evidence that a low nadir cell count compromised humoral responses evaluated six months after the second dose or one month after the third dose, nor did Gianserra et al. [20] or El Moussaoui et al. [21] after an interval of 4 weeks and 2–8 weeks, respectively, from the third dose. The different result we obtained could be explained by different kinetic models of humoral response used in these studies (our interval was the longest and estimated the half-life of anti-S antibodies as 55 days) [37], but caution is needed because of the small sample size.

Our study has three main limitations. Firstly, previous SARS COV-2 infection was excluded on clinical basis and not by testing anti N SARS-CoV-2–specific test. However, not only was this approach already used in other works [20], but also, a single anti-N evaluation could not be sufficient to exclude an exposition to the virus because of humoral clearance, and these antibodies are not produced by a significant portion of infected patients [38,39]. Unfortunately, patients with asymptomatic infection have low SARS-CoV-2 N antibody titers [40] and a possible infection occurring early in the pandemic could not be identified even in case of antibody testing. Furthermore, asymptomatic SARS-CoV-2 infection is common in patients with HIV infection [41] and it is possible that previously infected subjects were included, despite the criteria we applied to select only SARS-CoV-2 infection naïve individuals. Summing up, there could be uncertainty about whether natural infection could impact on IgG response at our study time points. Secondly, the results are not generalizable to all patients with HIV infection because all the subjects enrolled were on ART and had a high CD4+ cell count. Thirdly, by design, a control group of subjects negative for HIV infection was not included. It was not easy to enroll a cohort of healthy subjects including a comparable number of individuals (vaccinated with three doses of BNT162b2 mRNA vaccine and not with other authorized vaccines) who agreed to be sampled six months after the second and after the third vaccine dose. During the study period, patients with HIV infection were prioritized for vaccination, being considered a frail population, with respect to HIV-negative patients; therefore, this program and the relative sampling preceded a possible sampling of healthy subjects, especially the one after the third dose. Moreover, HIV-positive subjects underwent routine laboratory testing to monitor ART safety and efficacy; for these reasons, an additional investigation in these patients was more appropriate and ethical. On the other hand, the availability of IgG serum value distribution by quartiles in healthy subjects would have been extremely useful information for evaluating humoral response time course in HIV positive patients and validate our approach.

## 5. Conclusions

We demonstrated HIV patients’ persistence of anti-spike IgG, with high serum levels being demonstrated in most patients, six months after the third dose of the BNT162b2 mRNA vaccine and the predictive role of the humoral response after the second dose in patients with LLV and OV. Our results may help to formulate more stringent recommendations and motivate patients to accept further vaccine doses. Immunological controls are needed to check the persistence of responses and could help in designing vaccination strategies for selected patients, who could be better identified by multiple plasma HIV viremia tests. Further studies evaluating the IgG time course after the fourth dose or in the case of breakthrough COVID-19 may help elucidate the influence of HIV viremia detectability.

## Figures and Tables

**Figure 1 vaccines-11-00082-f001:**
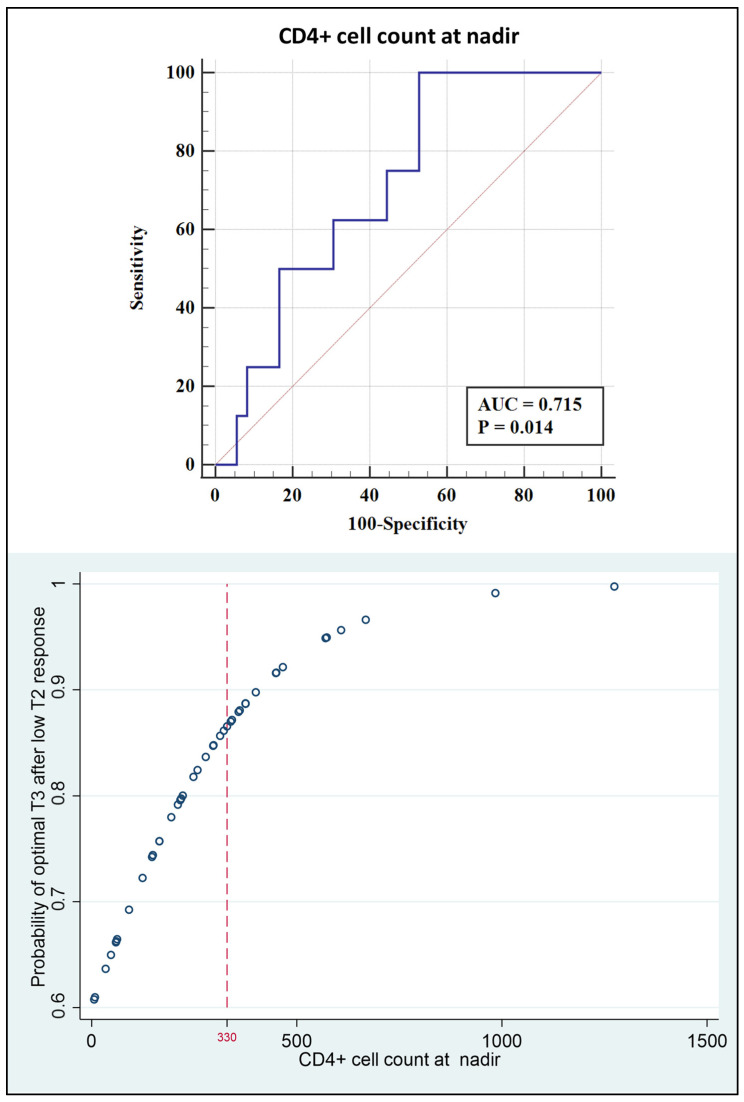
(**Upper**): Receiver operating characteristic curve analysis of the CD4+ count at nadir for low or intermediate IgG response at T3 in patients with low IgG response at T2. (**Lower**): Logistic model prediction of optimal T3 response probability among the same patients. AUC: area under the curve.

**Figure 2 vaccines-11-00082-f002:**
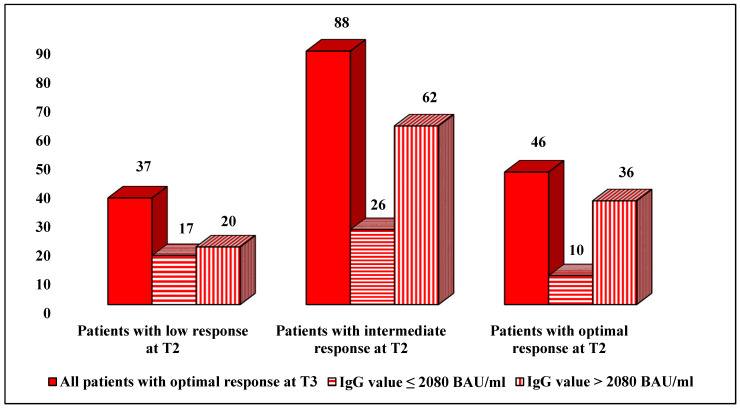
Description of the distribution of IgG values (values ≤ 2080 BAU/mL vs. values > 2080 BAU/mL) in patients with optimal response at T3.BAU: binding antibody units.

**Figure 3 vaccines-11-00082-f003:**
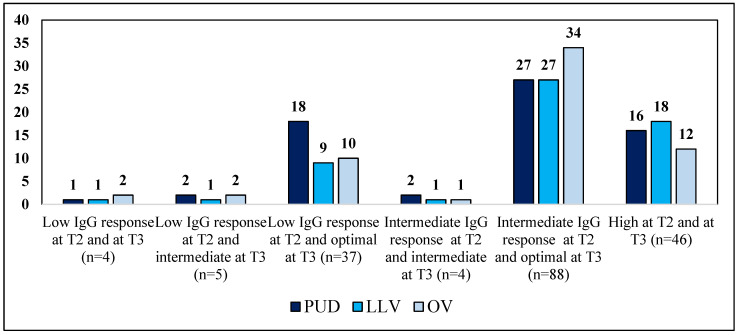
Description of the frequency of patients with PUD, LLV and OV in the different combinations of IgG response modification from T2 (6 months after the second vaccine dose) to T3 (six months after the third vaccine dose). PUD: plasma HIV viremia persistently undetectable; LLV: low-level viremia, no plasma HIV-RNA value was >20 copies/mL; OV: ongoing viremia, if at least one plasma value was >20 copies/mL.

**Table 1 vaccines-11-00082-t001:** Main characteristics of the study population.

	All the Patients (*n* = 184)
Male, *n* (%)	152 (82.6)
Age at HIV infection diagnosis (years) ^1^	38 (30–45)
Age at study enrollment (years) ^1^	55 (48–61)
Absolute number of CD4+ cell count at nadir (cells/mm^3^) ^1^	258 (107–384)
Overall CD4+ cell count (cells/mm^3^) ^1^	639 (486–845)
Delta CD4 cell count (cells/mm^3^) ^1^	372 (221–550)
Patients with persistently undetectable plasma HIV RNA, *n* (%)	66 (35.9)
Patients with low level plasma HIV RNA, *n* (%)	57 (31)
Patients with ongoing plasma HIV RNA, *n* (%)	61 (33.1)
Patients with negative serum IgG level at T2, *n* (%)	3 (1.6%)
Patients with serum IgG level >2080 BAU/mL, at T2 *n* (%)	16 (8.7)
Patients with negative serum IgG level at T3, *n* (%)	1 (0.5%)
Patients with serum IgG level >2080 BAU/mL at T3, *n* (%)	118 (64.1)

^1^ median and interquartile range. BAU: binding antibody units.

**Table 2 vaccines-11-00082-t002:** Absolute number of CD4+ cell count at nadir and as overall CD4+ cells value according to IgG response at T2 and plasma HIV RNA control.

Patients with Persistently Undetectable Viremia (*n* = 66)						
	Low Response (*n*,% = 21, 31.8)	Intermediate Response (*n*,% = 29, 44)	Optimal Response (*n*,% = 16, 24.2)	*p* L vs. I	*p* L vs. O	*p* I vs. O
CD4+ cell count at nadir (cells/mm^3^) ^1^	317 (157–425)	310 (157–428)	326 (134–408)	0.7145	0.7388	0.8852
Overall CD4+ cell count (cells/mm^3^) ^1^	575 (463–1051)	673 (508–812)	647 (552–841)	0.5229	0.9511	0.5533
Patients with low level viremia (*n* = 57)						
	Low response (*n*,% = 11, 19.3)	Intermediate response (*n*,% = 28, 49.1)	Optimal response (*n*,% = 18, 31.6)	*p* L vs. I	*p* L vs. O	*p* I vs. O
CD4+ cell count at nadir (cells/mm^3^) ^1^	321 (222–375)	320 (188–485)	222 (111–368)	0.9208	0.2918	0.1606
Overall CD4+ cell count (cells/mm^3^) ^1^	729 (475–1001)	721 (587–881)	662 (459–826)	0.6397	0.7419	0.3030
Patients with ongoing viremia (*n* = 61)						
	Low response (*n*,% = 14, 22.9)	Intermediate response (*n*,% = 35, 57.4)	Optimal response (*n*,% = 12, 19.7)	*p* L vs. I	*p* L vs. O	*p* I vs. O
CD4+ cell count at nadir (cells/mm^3^) ^1^	206 (60–322)	159 (76–302)	130 (45–214)	0.9367	0.3474	0.2447
Overall CD4+ cell count (cells/mm^3^)^1^	427 (269–814)	529 (400–928)	546 (412–569)	0.1120	0.8100	0.5582

L: low; I: intermediate; O: optimal. ^1^ median and range.

**Table 3 vaccines-11-00082-t003:** Logistic analysis of responsiveness to T2, overall and by viremic group.

Overall Responsiveness to T2 (N = 184)				
Variable	Odds Ratio	95% CI		*p*-value
CD4+ nadir count (cells/mm^3^)	0.99	0.99	1.01	0.153
CD4+ delta count (cells/mm^3^)	0.99	0.99	1.01	0.943
CD4+ overall count (cells/mm^3^)	0.99	0.99	1.01	0.355
Age on Enrollment (years)	1.04	0.98	1.1	0.159
Age on Diagnosis (years)	0.98	0.94	1.04	0.6
Responsiveness to T2 in persistently undetectable viremia group (N = 66)				
Variable	Odds Ratio	95% CI		*p*-value
CD4+ nadir count (cells/mm^3^)	0.99	0.99	1.01	0.7
CD4+ delta count (cells/mm^3^)	0.99	0.99	1.01	0.72
CD4+ overall count (cells/mm^3^)	0.99	0.99	1.01	0.78
Age on Enrollment (years)	1.02	0.97	1.01	0.33
Age on Diagnosis (years)	0.99	0.95	1.04	0.84
Responsiveness to T2 in low level viremia group (N = 57)				
Variable	Odds Ratio	95% CI		*p*-value
CD4+ nadir count (cells/mm^3^)	0.99	0.99	1.01	0.12
CD4+ delta count (cells/mm^3^)	1.01	0.99	1.01	0.73
CD4+ overall count (cells/mm^3^)	0.99	0.99	1.01	0.35
Age on enrollment (years)	0.98	0.93	1.01	0.52
Age on diagnosis (years)	0.95	0.89	1.01	0.11
Responsiveness to T2 in ongoing viremia group (N = 61)				
Variable	Odds Ratio	95% CI		*p*-value
CD4+ nadir count (cells/mm^3^)	0.99	0.99	1.01	0.21
CD4+ delta count (cells/mm^3^)	0.99	0.99	1.01	0.9
CD4+ overall count (cells/mm^3^)	0.99	0.99	1.01	0.39
Age on enrollment (years)	1.03	0.96	1.01	0.37
Age on diagnosis (years)	1.1	0.01	1.52	0.11

**Table 4 vaccines-11-00082-t004:** Description of longitudinal evolutions of humoral response at T3 according to response at T2. Data are expressed as absolute number and as percentage respect to the relative frequency. The same quartile limits in BAU/mL were applied both at T2 and at T3.

	Patients with Low Response at T3 *n* (%)	Patients with Intermediate Response at T3 *n* (%)	Patients with Optimal Response at T3 (%)
Patients with low response at T2, *n* = 46	4 (8.7)	5 (10.9)	37 (80.4)
Patients with intermediate response at T2, *n* = 92	0	4 (4.3)	88 (95.7)
Patients with optimal response at T2, *n* = 46	0	0	46 (100)

**Table 5 vaccines-11-00082-t005:** Logistic analysis of responsiveness to T3, overall and by viremic group. In bold: results with statistical significance level *p* < 0.05.

Overall Responsiveness to T3				
Variable	Odds Ratio	95% CI		*p*-value
Responsiveness to T2	3.23	1.5	6.9	0.003
CD4+ nadir count (cells/mm^3^)	1.002	0.99	1.01	0.62
CD4+ delta count (cells/mm^3^)	1.01	0.99	1.01	0.26
CD4+ overall count (cells/mm^3^)	1.01	0.99	1.01	0.31
Age on enrollment (years)	1.04	0.98	1.1	0.16
Age on diagnosis (years)	0.98	0.94	1.03	0.6
Responsiveness to T3 in persistently undetectable viremia group				
Variable	Odds Ratio	95% CI		*p*-value
Responsiveness to T2	2.16	0.79	5.86	0.09
CD4+ nadir count (cells/mm^3^)	0.99	0.99	1.01	0.79
CD4+ delta count (cells/mm^3^)	1.002	0.99	1.01	0.34
CD4+ overall count (cells/mm^3^)	1.001	0.99	1.003	0.82
Age on enrollment (years)	1.001	0.91	1.09	0.93
Age on diagnosis (years)	0.99	0.92	1.06	0.79
Responsiveness to T3 in low level viremia group				
Variable	Odds Ratio	95% CI		*p*-value
Responsiveness to T2	5.23	0.75	36.68	0.021
CD4+ nadir count (cells/mm^3^)	1	0.99	1	0.31
CD4+ delta count (cells/mm^3^)	1.001	0.99	1.01	0.96
CD4+ overall count (cells/mm^3^)	1.003	0.99	1.01	0.2
Age on enrollment (years)	1.09	0.95	1.24	0.2
Age on diagnosis (years)	1.08	0.94	1.25	0.24
Responsiveness to T3 in ongoing viremia group				
Variable	Odds Ratio	95% CI		*p*-value
Responsiveness to T2	4.27	1.01	18.12	0.013
CD4+ nadir count (cells/mm^3^)	1	0.99	1.01	0.45
CD4+ delta count (cells/mm^3^)	1.002	0.99	1.01	0.45
CD4+ overall count (cells/mm^3^)	1.001	0.99	1.01	0.78
Age on enrollment (years)	1.05	0.005	78.31	0.24
Age on diagnosis (years)	0.94	0.87	1.02	0.17

## Data Availability

The raw data on demographics and clinical status of participants are protected and not available due to data privacy laws. The processed data are available by the corresponding author upon reasonable request.

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
