# Peer review of "Humoral Response after Two Doses of BNT162b2 mRNA Vaccine Has a Role in Predicting Response after Three Doses That Is Related to Plasma HIV Viremia and Nadir CD4+ Cell Count in HIV-Positive Patients"

_vaccines, 2022, doi:10.3390/vaccines11010082_

Round 1

Reviewer 1 Report

1. The authors aimed to not to include patients with previous SARS-CoV-2 infection however there is no test that demonstrate that the patients in the study had no anti-SARS-CoV-2 antibodies prior to vaccination to make sure that they did not have  a previous infection!

2. The result section can have separated paragraphs with title to make it easier to read

3. The authors did not include healthy controls for comparison. This makes many results hard to interpret especially their IgG level (intermediate, etc.)

4. There are a lot of English issues that make the manuscript hard to understand.

5. when were the total and nadir CD4 count measured?

Reviewer 2 Report

The article has explained the effect of HIV on the Covid19 vaccine mediated antibody responses. The study design is proper. I have one concern,

The author needs to explain the criteria used to select 194 patients for this study. 

Author Response

The article has explained the effect of HIV on the Covid19 vaccine mediated antibody responses. The study design is proper. I have one concern,

Point 1: The author needs to explain the criteria used to select 194 patients for this study.

Response 1: all the HIV patients vaccinated with three doses of BNT162b2 mRNA vaccine, who had SARS-CoV-2 serology (IgG antibodies against a trimeric S-protein antigen) tested and no prior laboratory-confirmed COVID-19 or anamnestic data suggesting a possible SARS-CoV-2 infection not further evaluated were included in the study.

We agree with the Reviewer about the need of greater clarity and we modified the text as follows: “No other exclusion criteria was applied.”

Reviewer 3 Report

The manuscript of Basso M et al. presents a cohort of HIV-positive adult patients who received the anti-SARS-Cov-2 vaccine in 2021. They characterize clinical and laboratory manifestations and attempt to evaluate predictors of the humoral immune response.

This study is attractive, mainly due to the group of patients evaluated with a previous immune depression such as VIH infection. In addition, the design of the study is appropriate and well-performed.

However, the statistical analysis requires several corrections to interpret this study appropriately. Here I detail the concerns and suggestions:

  1. This study MUST be analyzed by a biostatistician with experience in observational clinical studies. He/she should evaluate this manuscript's original analysis and perform appropriate statistical analysis.
  2. The study has a relatively small number of patients to perform ROC analysis. The biostatistician must determine the most appropriate analysis that can be performed using available data.
  3. Most ROC analyses show non-statistically differences between analyses. This is expected due to the sample size, i.e., a potential type II statistical error. A power analysis is suggested if the authors consider this analysis can be performed. Please add the statistical justification for this analysis in Methods and Discussion if an appropriate ROC analysis can be performed.
  4. A potential analysis that can be performed is a logistic or linear regression analysis to evaluate potential predictors of the immune response. Although this analysis is inferior to ROC analysis related to the interpretation and potential clinical relevance of these results, it can be performed using your data and current sample size. A potential article that can be useful is (Clavero R et al. Vaccines 2022), which evaluated a cohort of renal patients with a similar sample size as this study.

I expect a new version of the manuscript with appropriate analysis, which can be accepted for publication in the Vaccines journal.

Round 2

Reviewer 1 Report

The authors must add ore details about the limitations because of the lack of healthy control and the previous antibody levels in the manuscript!

Author Response

Point 1. The authors must add more details about the limitations because of the lack of healthy control and the previous antibody levels in the manuscript
Response 1: Thanks for the opportunity to ameliorate our work. We are aware that the lack of a control group of healthy subjects and the unavailability of serology testing to exclude a previous SARS-CoV-2 infection are important limits of our study. The following test was added to detail the limitations: “It was not easy to enroll a cohort of healthy subjects including a comparable number of individuals (vaccinated with three doses of BNT162b2 mRNA vaccine and not with other authorized vaccines) who accepted to be sampled six months after the second and after the third vaccine dose. During the study period patients with HIV infection were prioritized for vaccination, being considered a frail population, with respect to HIV negative patients; therefore this program and the relative sampling preceded a possible sampling of healthy subjects, especially the one after the third dose. Moreover, HIV positive subjects underwent a routine laboratory testing to monitor ART safety and efficacy: for these reasons an additional investigation in these patients was more appropriate and ethical. On the other hand, the availability of IgG serum value distribution by quartiles in healthy subjects would have been an extremely useful information for evaluating humoral response time course in HIV positive patients and validate our approach”. “Furthermore, asymptomatic SARS-CoV-2 infection is common in patients with HIV infection [Overton et al] and it is possible that previously infected subjects were included, despite the criteria we applied to select only
naïve individuals. Summing up, there could be uncertainty about whether natural infection could impact on IgG response at our study time points”.

Reviewer 3 Report

The authors have completed all the requests I performed.

I consider that the article is suitable for publication in Vaccines journal.

Author Response

Point 1. The authors have completed all the requests I performed. I consider that the article is suitable for publication in Vaccines journal.
Response 1: We thank the Reviewer for the careful evaluation of our study and for the precious e accurate indications which allowed to increase the quality of the paper.